# Coverage-guided differential testing of TLS implementations based on syntax mutation

**Yan Pan** [ID], **Wei Lin, Yubo He, Yuefei Zhu**\*

State Key Laboratory of Mathematical Engineering and Advanced Computing, Zhengzhou, China

\* yfzhu17@sina.com

**Data Availability Statement:** All source code files are available from https://gitee.com/z11panyan/CGDTSM.

**Funding:** This work is supported by National Key Research and Development Project of China

## Abstract

Transport layer security (TLS) protocol is the most widely used security protocol in modern network communications. However, protocol vulnerabilities caused by the design of the network protocol or its implementation by programmers emerge one after another. Meanwhile, various versions of TLS protocol implementations exhibit different behavioral characteristics. Researchers are attempting to find the differences in protocol implementations based on differential testing, which is conducive to discovering the vulnerabilities. This paper provides a solution to find the differences more efficiently by targeting the TLS protocol handshake process. The differences of different implementations during the fuzzing process, such as code coverage and response data, are taken to guide the mutation of test cases, and the seeds are mutated based on the TLS protocol syntax. In addition, the definition of duplicate discrepancies is theoretically explored to investigate the root cause of the discrepancies and to reduce the number of duplicate cases that are caused by the same reason. Besides, the necessary conditions for excluding duplicate cases are further analyzed to develop the deduplication strategy. The proposed method is developed based on open-source tools, i.e., NEZHA and TLS-diff. Three types of widely used TLS protocol implementations, i.e., OpenSSL, BoringSSL, and LibreSSL, are taken for experimental testing. The experimental results show that the proposed method can effectively improve the ability to find differences between different implementations. Under the same test scale or the same time, the amount of discrepancies increases by about 20% compared to TLS-diff, indicating the effectiveness of the deduplication strategy.

## Introduction

With the rapid development of computer networks, more and more applications are being transformed into network applications. As carriers of various network transmissions, network protocols occupy an important position in the entire network and play an essential role in ensuring secure communication between network devices. The logical errors in network protocol design and the bugs implemented by programmers in programming lead to the fragility of protocol implementation. As a widely used encryption protocol, TLS protocol is fundamental to encrypted communication. Due to various versions of TLS protocol, different

(2019QY1300). We declare that we have no
financial and personal relationships with other
people or organizations that can inappropriately
influence our work, there is no professional or
other personal interest of any nature or kind in any
product, service or company that could be
construed as influencing the position presented in,
or the review of, the manuscript.

**Competing interests:** NO authors have competing
interests.

vulnerabilities are introduced by different implementations, such as the heart drip vulnerability of OpenSSL, CCS (Change Cipher Spec) injection vulnerability, and the goto fail vulnerability of GnuTLS.

To analyze the vulnerability of TLS implementation, researchers applied different analysis methods for different processes, such as source code analysis, fuzzing, and formal methods [1]. For standardized verification of the protocol implementation [2, 3], Chaki et al. [4] combined software model detection with standard protocol security models to automatically analyze the authentication and confidentiality of the protocol in the C language. For the protocol state machine [5, 6], Ruiter et al. [7] modeled a state machine for implementing the TLS protocol based on the active learning method. They also manually analyzed the generated state machine to find logical vulnerabilities. For data interaction, Somorovsky [8] developed an open-source framework called TLS-Attacker, which can perform fuzzy testing of the processing of data interaction over the TLS protocol. Concerning certificate validation, Brubaker et al. [9] performed differential testing of the certificate validation process in various implementations.

Among the existing methods, differential testing originates from software regression testing. It tests different versions of implementations by inputting semi-valid test cases and analyzing the differences between different implementations. Addressing the problem of poor performance in test case generation, Petsios et al. [10] introduced the concept of $\delta$ diversity. They integrated differential testing with guided testing ideas and developed the NEZHA open-source platform to test the consistency of behavior across multiple test programs. In addition, they heuristically proposed the concept of a path combination, combining the execution paths of multiple implementations into path combinations. They suggested that the test case is useful for finding differences in the output if it is a new combination of paths and even if the path of one of implementation is a covered path. TLS-diff [11] applied differential testing to the handshake process of TLS and proposed stimulating multiple TLS implementations with equivalent inputs via semi-randomly generated TLS protocol messages. Also, the implementation errors were analyzed based on the differences in their response. This method analyzed the interaction of the first packet data in the handshake process but lacked guidance.

For the TLS handshake process, the mutation efficiency is reduced due to the independence of the domain of NEZHA and the strong structure of the data packet. To more effectively detect differences in TLS implementation during the handshake process, this paper proposes a coverage-guided differential testing method based on syntax mutation. Given the TLS handshake protocol, this method improves the applicability of the NEZHA based algorithm and replaces the original random mutation with syntax-based mutation when generating test cases to improve efficiency.

Meanwhile, since the implementations of the above two methods do not give additional judgments about the results of the difference, that is, the difference can be caused by the same factor. Therefore, the "duplication" is defined in this paper based on the root cause of the alarm. Besides, the necessary conditions are discussed theoretically and implemented in the tool so that the repetitive difference test cases can be eliminated to some extent.

The main contributions of this paper are summarized as follows:

- A hybrid methodology composed of coverage guidance and syntax mutation for differential testing on the first interaction of TLS protocol is proposed.

- The method of eliminating duplicate discrepancies based on code coverage is discussed.

- To facilitate the work of other researchers, the modified tool is open-sourced and provided at https://gitee.com/z11panyan/CGDTSM.

The rest of this paper is organized into six parts. The first part introduces the related work; the second part discusses related research; the third part describes the proposed method; the fourth part introduces experiment and evaluation; the fifth part discusses the manual analysis of differential testing cases; and the sixth part is a conclusion and perspective.

## Related work

The related work is mainly presented from two aspects: fuzzing and differential testing.

### Fuzzing

Fuzzing is currently a research hotspot for finding software vulnerabilities. Abnormal samples are generated and sent to the testing software for execution, so that the deviations in program processing can be detected and their vulnerabilities analyzed. Test cases generation and the control strategy feedback data are two key components. Fuzzing is usually divided into black-box, gray-box, and white-box testing based on the feedback data provided by the executing program. Black-box testing does not require any feedback data from the target program and pays more attention to mutation methods, such as input structure-based mutation strategies [12, 13], and input structure-based generation strategies built on deep learning [14–16]. White-box testing is based on the internal logic of the program to create test cases using dynamic symbolic execution and heuristic search algorithms for maximum coverage. SAGE [17] is a typical tool, but the prerequisites and complexity of white box testing are relatively high. Grey-box testing mainly focuses on the code coverage (basic blocks, paths, functions) and data flow. Common tools such as AFL [18] and LibFuzzer [19] are both used to obtain code coverage with code instrumentation (source code, binary). AFLNET [20] uses state-feedback and coverage-feedback to guide the mutation of seeds and treat the message sequences as the fuzzing input to enable deep interaction with protocol implementations. And the work in [21] focuses on data flow at runtime.

On Usenix2015, Ruiter et al. modeled for the first time a state machine of the TLS protocol implementations based on the active learning method. They also manually analyzed the generated state machine to find logical vulnerabilities. Besides, they improved the W-method equivalent query algorithm [22] based on the LearnLib framework [23], which reduces the number of equivalent queries and speeds up the construction of the state machine. According to the above method, state machines implemented by the TLS protocol can be built quickly. The state machines can be manually inspected to find incorrect state transitions or redundant states. Compared with real source code, bugs in the implementation can be fixed. Somorovsky proposed an open-source framework called TLS-attacker for evaluating the security of TLS implementations, which modified the source code of TLS client to make all protocol fields variable. At run time, the script can use the specified fuzzing operation to mutate, create test cases, and test the TLS implementation.

### Differential testing

Differential testing was first proposed by Evans [24] to analyze the difference between old and new versions of software. Since there are various implementations of TLS, Brubaker et al. introduced the idea of differential testing into the certificate verification process in the SSL implementation. They collected 243,246 certificates over the network and generated the set of certificates by randomly changing the fields of the certificate. It can effectively detect differences in the certificate validation process across different implementations. Similarly, focusing on the certificate generation, Chen [25] diversified seed certificates by adapting Markov Chain Monte Carlo sampling and Tian [26] assembled the certificates based on the standard RFC

(Request for Comments), which both improve traditional differential testing. HVLearn [27] analyzed host name validation when verifying certificates. SFADiff [28] combines automatic inference with differential testing. It derived the symbolic finite automata model by querying the target program using the black box method, and checked for differences in the inferred model. Walz et al. introduced the black box feedback idea of NEZHA to the TLS-diff [29]. They proposed a response-distribution guided strategy that uses a certain probability as a seed for a mutation to generate new test cases. This method depends on the NEZHA, so it matches the black box test.

In addition, differential testing has also been applied to other areas, such as DIFFUZZ for side-channel analysis [30], DLFuzz for deep learning systems [31], processing differences in malware recognition tools [32], and in combination with symbolic execution for further broaden observed differences [33]. Basically, differential testing and fuzzing create automatic or semi-automatic data as input to the program, and frack deviations in the program. Therefore, structured mutation and guided strategy are very important.

## Methodology

### Motivation

The SSL protocol includes two protocol sublayers. The bottom layer is the SSL recording protocol layer; the upper layer is the SSL handshake protocol layer. The SSL handshake protocol layer includes the SSL handshake protocol, the SSL cipher change specification protocol, and the SSL alert protocol. This paper focuses on the first communication in the SSL handshake protocol. After receiving the ClientHello packet, the server will parse the payload according to the grammar and validate each attribute field. If these attribute fields comply with the standard, the handshake sub-protocol is invoked with a value of 22 in the identification field; otherwise, the alarm sub-protocol is invoked with a value of 21 in the identification field.

Based on the structure of ClientHello data packets, TLS-diff proposes the concept of generic message trees, which is an ordered rooted tree, with each node representing a specific message field. A leaf node in a tree is an atomic message field (for example, an integer field) that can be directly transformed from the original data. An internal node in the tree is a compound message field, and the contents in that field are recursively retrieved from its child nodes. TLS-diff first converts the original data packet into a general message tree based on certain conversion rules; then it mutates the nodes of the tree with the following eight types of mutation operations: $O_{void}, O_{rem}, O_{dupl}, O_{trunc}^{fuzz}, O_{int}^{fuzz}, O_{cont}^{fuzz}, O_{app}^{fuzz}, O_{syn}^{fuzz}$. Finally, the mutated tree structure is converted into a data packet through a serialization process. The tool generates many ClientHello data packets based on this mutation strategy and inputs them to different protocol implementations to obtain different responses.

NEZHA combines the idea of coverage guidance and guides and optimizes the creation of use cases in the fuzzing process. It focuses on the difference in the path and the difference in the output of different implementations. For a specific test case, the execution path and output of each test program are the combination of the execution paths and the output combination of the test case. If a combination of the execution path or an output combination is recreated, it is considered significant for finding output differences. In this case, the use case is added to the original dataset and used as the raw data for subsequent mutation. The original implementation of NEZHA provides three strategies: the path $\delta$ diversity (fine) matches a combination of execution paths: the path $\delta$ diversity (coarse) matches a combination of the execution path numbers: and output $\delta$ diversity matches the output combination. Meanwhile, if there is a discrepancy in the output of the test application, NEZHA will add the corresponding input to the general set of differences.

The above two methods have the same problem, if the responses of two differences caused by different reasons are the same, such as the tuple <21, 21, 0>, one of them will be discarded, leading to the loss of valid discrepancies. However, if the discrepancies with the same response are not discarded, but are recorded as the final discrepancies, the number of discrepancies in the literature [29] can reach thousands of levels. Therefore, it is difficult to judge the reliability, and the manual analysis is expensive. In addition, differential testing for certificate verification is facing the same problem [25, 26].

As for the generation of test cases, TLS-diff is a black box test based on grammatical mutation. While the generated test samples may closely match the grammar of ClientHello data packets, the mutation is relatively blind. The original intention of NEZHA is to conduct domain-independent guided testing, and the mutation strategy is random. However, there are still too many invalid use cases, hindering the improvement of efficiency.

## Algorithm design

Regarding the above issues, this article proposes a hybrid technique called CGDTSM (Coverage-guided Differential Testing with Syntax-based Mutation) that applies the syntax-based mutation strategy proposed by TLS-diff to NEZHA. While it disrupts the domain-independent characteristics of NEZHA, it can amplify the mutation of NEZHA's TLS packet to a certain extent.

At the same time, to minimize the "duplication" of test cases as much as possible, the test cases deduplication strategy is discussed below. Table 1 shows the relevant symbols often used in the process.

Firstly, the "duplicate difference" is defined based on underlying definitions.

**Definition 1**. A set of commands consisting of assembly instructions is denoted as

$$\mathcal{I} = \{I_i | \ i = 1, \ldots, n\}.$$

**Definition 2**. Program $p$ is an ordered set of instructions, denoted as

$$p = \{I_{i_1}, I_{i_2}, \ldots, I_{i_t} | I_{i_t} \in I\}.$$

The set of programs under test is denoted as $\mathcal{P} = \{p_i | \ i = 1, \ldots, m\}$. These programs are various implementations of TLS.

**Definition 3**. A test case denoted as $c$ is the single input generated for a specific type of program. The set of test cases is denoted as $\mathcal{C} = \{c_i \ | \ i = 1, \ldots, k\}$.

**Definition 4** (Instruction execution). Given a program $p \in \mathcal{P}$, a test case $c \in \mathcal{C}$, the execution of an instruction is defined as a mapping $e : \mathcal{C} \times \mathcal{I} \times \mathcal{T} \to \{true, false\}$. Since the program selectively executes different instructions at runtime, the symbol $t$ is introduced to represent a particular execution process. The set consisting of the symbols $t$ is denoted as $T$. If the instruction $I \in p$ is executed by a test case $c$, then $e(c, I, t) = true$; otherwise, $e(c, I, t) = false$.

**Definition 5** (Basic block). For a given program $p \in \mathcal{P}$, a set of basic blocks $\mathcal{B}_p = \{b_i | \ i = 1, \ldots, l\}$ is uniquely determined, where the basic block $b \in \mathcal{B}_p$ is a sequence of statements with atomicity, and can be represented as an ordered set of one or more instructions: $b = \{I_{i_s}, I_{i_{s+1}} \ldots, I_{i_{s+j}} | I_{i_{s+j}} \in \mathcal{I}\}.$

**Table 1. Description of used notation.**

| Symbols | $\mathcal{I}, I$ | $\mathcal{P}, p$ | $\mathcal{C}, c$ | $\mathcal{B}, b$ | $\mathcal{A}, a$ |
|---------|------|------|------|------|------|
| Meanings | Instructions | Programs | Test cases | Basic blocks | Alarm basic blocks |

When $p = \{I_{i_1}, \ldots I_{i_{s-1}}, I_{i_s}, \ldots, I_{i_{s+j}}, I_{i_{s+j+1}} \ldots, I_{i_t} | I_{i_t} \in \mathcal{I}\}$ is selected, the basic block is $b = \{I_{i_s}, I_{i_{s+1}} \ldots, I_{i_{s+j}} | I_{i_{s+j}} \in \mathcal{I}\}$ if and only if:

① $\forall c \in \mathcal{C}, t \in \mathcal{T}$, s.t. $e(c, I_{i_s}, t) = e(c, I_{i_{s+1}}, t) = \ldots = e(c, I_{i_{s+j}}, t)$

② $\exists c \in \mathcal{C}, t \in \mathcal{T}$, s.t. $e(c, I_{i_{s-1}}, t) \neq e(c, I_{i_s}, t), e(c, I_{i_{s+j}}, t) \neq e(c, I_{i_{s+j+1}}, t)$

Based on additional definitions, the mapping $e$ can be extended from the execution of instructions to the basic block. The definition is as follows.

**Definition 6**. Given $p \in \mathcal{P}, c \in \mathcal{C}$, the mapping is $e : \mathcal{C} \times \mathcal{B}_p \times \mathcal{T} \to \{true, false\}$. If the basic block $b \in \mathcal{B}_p$ is executed, then $e(c, b, t) = true$, otherwise $e(c, b, t) = false$.

**Definition 7**. Given $p \in \mathcal{P}, c \in \mathcal{C}$, the set of covering basic blocks in an execution can be denoted as $\mathcal{B}_p(c, t) = \{ b \in \mathcal{B}_p | e(c, b, t) = true \}$, and the basic blocks covered by the test case $c$ can be denoted as $\mathcal{B}_p(c) = \cup_{t \in \mathcal{T}} \mathcal{B}_p(c, t)$. The space formed by $\mathcal{C}, \mathcal{P}, \mathcal{B}_p(c)$ can be denoted as $\mathfrak{B} = \{\mathcal{B}_p\{c\} \mid p \in \mathcal{P}, c \in \mathcal{C}\}$.

**Definition 8**. Define a mapping $f : \mathcal{C} \times \mathcal{P} \to \mathfrak{B}$ for a given $p \in \mathcal{P}, c \in \mathcal{C}, f(p, c) = \mathcal{B}_p(c)$.

The related definitions which are used in this paper are introduced below.

**Definition 9** (Response output). In this paper, the set of programs to be tested $P$ is composed of TLS implementations, and the output of programs to be tested can be defined as a mapping $out : \mathcal{C} \times \mathcal{P} \to \{21, 0\}$. For $p \in \mathcal{P}, c \in \mathcal{C}$,

$$out(p, c) = \begin{cases} 21, & if\ Alert \\ 0, & if\ Handshake \end{cases}.$$

The set of programs $p$ generating an alarm for the test case $c \in \mathcal{C}$ is denoted as $\mathcal{P}_a(c) = \{p \in \mathcal{P} \mid out(p, c) = 21\}$.

**Definition 10** (Alarm basic block). Given $p \in \mathcal{P}$, the class of basic block sets $\mathcal{A}_p, a \in \mathcal{A}_p$ is determined uniquely if and only if

① $\forall\ t \in \mathcal{T}, \quad \forall\ c \in \mathcal{C}, s.t.\ out(p, c) = 0, e(c, a, t) = false$;

② $\exists\ c \in \mathcal{C}, s.t.\ out(p, c) = 21, e(c, a, t) = true$.

It is called an alarm basic block.

**Property 1**. Given $p \in \mathcal{P}$ and $c \in \mathcal{C}$, if $out\ (p, c) = 21$, then $\forall\ t_1, t_2 \in \mathcal{T}, \forall\ a \in \mathcal{A}_p$.
If $a \in \mathcal{B}_p(c, t_1)$, then $a \in \mathcal{B}_p(c, t_2)$; if $out(p, c) = 0, \forall\ a \in \mathcal{A}_p, \forall\ t \in \mathcal{T}, a \notin \mathcal{B}_p(c, t)$.

During program execution, the main blocks are executed in a specific order. This paper focuses on the first alarm basic block during running.

**Definition 11**. Given $p \in \mathcal{P}$ and $c \in \mathcal{C}$, the first alarm base block is defined as a mapping $h : \mathfrak{B} \to \mathcal{A} \cup \{0\}$,

$$h\Big(\mathcal{B}_p(c)\Big) = \begin{cases} a, & if\ out(p, c) = 21 \\ 0, & if\ out(p, c) = 0 \end{cases}.$$

In this paper, the basic block $a$ is the cause of alarm. Based on Quality 1, for given $p \in \mathcal{P}$ and $c \in \mathcal{C}, \forall t_1, t_2 \in \mathcal{T}$, it holds $h(\mathcal{B}_p(c, t_1)) = h(\mathcal{B}_p(c, t_2)) = h(\mathcal{B}_p(c))$.

**Definition 12**. Define a composite mapping $g$: $h \circ f$, and $g : \mathcal{C} \times \mathcal{P} \to \mathcal{A} \cup \{0\}$

$$g(p, c) = \begin{cases} a, & if \ Alert \\ 0, & if \ Handshake \end{cases}.$$

**Definition 13**. Given $p \in \mathcal{P}$, a relation on $\mathcal{C}$ is defined as

$$\approx_g = \{< c_1, c_2 > | c_1, c_2 \in \mathcal{C} \wedge g\{p, c_1\} = g\{p, c_2\}\}.$$

It is easy to prove that $\approx_g$ is an equivalent relation, and its equivalence class is denoted as $[c]_{\approx_g}$. In other words, if the elements in the equivalence class generate an alarm, the cause of the alarm is the same. This paper discusses only alarm test cases and the test cases with successful handshake are considered in the same way. Therefore, the elements in the equivalence class are called duplicate cases.

**Theorem 1**. Given $p \in \mathcal{P}, \forall \ c_1, c_2 \in \mathcal{C}, \exists t_1, t_2 \in \mathcal{T}$, if $\mathcal{B}_p(c_1, t_1) = \mathcal{B}_p(c_2, t_2)$, then $c_1 \approx_g c_2$.

*Proof*. If $\mathcal{B}_p(c_1, t_1) = \mathcal{B}_p(c_2, t_2)$, we have $h(\mathcal{B}_p(c_1, t_1)) = h(\mathcal{B}_p(c_2, t_2))$ from Definition 11. Further, we have $h(\mathcal{B}_p(c_1)) = h(\mathcal{B}_p(c_2))$. Then based on Definition 8 and 12, $g(p, c_1) = g(p, c_2)$, which means $c_1 \approx_g c_2$.

According to Theorem 1, if the sets of basic blocks which are triggered by two test cases are the same, the two test cases are considered a duplicate. When a test case is input into multiple program implementations at the same time, the following definition is added.

**Definition 14**. For a set of programs under test $\mathcal{P}_t \subseteq \mathcal{P}$, the test case $c$ is a different case, if and only if $\exists \ p_i, p_j \in \mathcal{P}_t, out(p_i, c) \neq out(p_j, c)$. The set of different cases is recorded as $C_{\mathcal{P}_t} = \{c \mid out(p_i, c) \neq out(p_j, c), \mid p_i, p_j \in \mathcal{P}_t\}$.

**Theorem 2**. Given a set of programs $\mathcal{P}_t, \approx_g$ is an equivalence relation on $\mathcal{C}_{\mathcal{P}_t}$.

**Definition 15**. For a set of tested programs $\mathcal{P}_t \subseteq \mathcal{P}, c_1, c_2 \in \mathcal{C}_{\mathcal{P}_t}$ are repeated test cases for this set if and only if $\mathcal{P}_a(c_1) = \mathcal{P}_a(c_2)$, and for $\forall \ p \in \mathcal{P}_t, c_1 \approx_g c_2$.

**Theorem 3**. For $\forall \ c_1, c_2 \in \mathcal{C}_{\mathcal{P}_t}, \exists t_1, t_2 \in \mathcal{T}$, if $\mathcal{P}_a(c_1) = \mathcal{P}_a(c_2)$ and $\forall \ p \in \mathcal{P}_a(c_1)$, $\mathcal{B}_p(c_1, t_1) = \mathcal{B}_p(c_2, t_2)$, then $c_1, c_2$ is the repeated test case for this set.

Theorem 3 is easy to prove. According to Theorem 3, the following deduplication method is formulated in this paper: for the difference case obtain the code coverage data of the program responding to the alert, and calculate its hash value. If it matches an existing hash library, then it is considered to satisfy the condition of Theorem 3. In other words, there is a difference case in the library that repeats with the test case, and it will not be put into the library, otherwise, it will be placed in the library.

Of course, the above definition can be extended to the verification class program, that is, a program that validates the input. A basic block in which an input is considered invalid is regarded as an alarm base unit that has the same property. Therefore, this paper describes the $\delta$ diversity of NEZHA in accordance with the above definition.

For the selected program $p_1, p_2 \in \mathcal{P}_t$, a certain alarm basic block can be denoted as $\mathcal{A}_{p_1} = \{a_{11}, a_{12}, \ldots, a_{1u}\}, \mathcal{A}_{p_2} = \{a_{11}, a_{12}, \ldots, a_{1v}\}, \exists \ c_1, c_2 \in \mathcal{C}$, which holds $e(c_1, a_{11}, t) = true$, $e(c_1, a_{21}, t) = true$, $e(c_2, a_{12}, t) = true$ and $e(c_2, a_{22}, t) = true$. Traditional coverage-guided strategy marks $c_1, c_2$ as meaningful test cases. If there is $c_3 \in \mathcal{C}$, s.t. $e(c_3, a_{11}, t) = true$ and $e(c_3, a_{22}, t) = true$, the traditional coverage guidance strategy does not mark, while the strategy of NEZHA considers the new basic block coverage to be a combination $(a_{11}, a_{22})$ generated by the test case $c_3$, which has a specific value for creating new output differences or anomalies.

Combined with the above definition, the CGDTSM algorithm proposed in this article is shown in Algorithm 1. $\mathcal{C}$ is the set of test cases, and $\mathcal{P}$ is the set of protocol implementations.

Lines 3–4 randomly select the test cases in the set and mutate based on the syntax, so that new test cases can be generated; lines 5–7 send each test case to several programs and record the execution path and response of several program; lines 8–10 indicate that if a new pattern is generated, the test case will be added to set $\mathcal{C}$ and the increased number of features will be used as the weight of the test case. Lines 11–17 eliminate the repeated cases according to the above deduplication algorithm and only record the discrepancies after deduplication. Lines 4, 12, and 13 are the key new enhancements on top of NEZHA.

**Algoritm 1** DiffTest: Report all discrepancies across applications $\mathcal{P}$ after n generations, starting from a corpus $\mathcal{C}$.

```
Input: The initial set of test cases C; The set of programs under test
P; The number of execution n;
Output: The set of difference cases diff;
1: procedure DiffTest(C, P, n)
2:   while generation ≤ n do
3:     input = RandomChoice(C)
4:     testcase = MutateOnSyntax(input)
5:     for app ∈ P do
6:       path, outputs = RUN(app, testcase)
7:     end for
8:     if NewPattern(testcase) then
9:       C = C ∪ testcase
10:     end if
11:     if IsDiscrepancy(testcase)
12:       hash = Hash(GetBitmap())
13:       if Hashmap.insert(hash,true)
14:         diff ∪ = testcase
15:         C = C ∪ testcase
16:       end if
17:     end if
18:   end while return diff;
```

## System design

The system design structure is shown in Fig 1. Based on the NEZHA framework (https://github.com/nezha-dt/nezha/), the shadow parts of the diagram have been modified, which correspond to Lines 4 and 12 in Algorithm 1 respectively. The mutation component invokes TLS-diff mutation strategy based on syntax. The feedback focuses on the output and coverage of basic blocks. If any condition is met, it is added to the source library.

To adapt to the latest SSL source code, the project and the SSL source code are compiled with Clang9 and instrumented based on Sanitizer Coverage.

## Evaluation

### Experimental design

Three of the most commonly used SSL implementations are used in the experiment, including BoringSSL, LibreSSL, and OpenSSL. Versions used are BoringSSL 2883, and master(3743aaf), LibreSSL 2.4.0 and 3.2.1, OpenSSL1.0.2(12ad22d), OpenSSL1.1.0(a3b54f0) and 3.0.0-alpha7--dev(b0614f0). The first is the version accepted in literature [10, 11], and the second is the latest version. NEZHA and TLS-diff are used as comparison methods to compare the effects of the CGDTSM method.

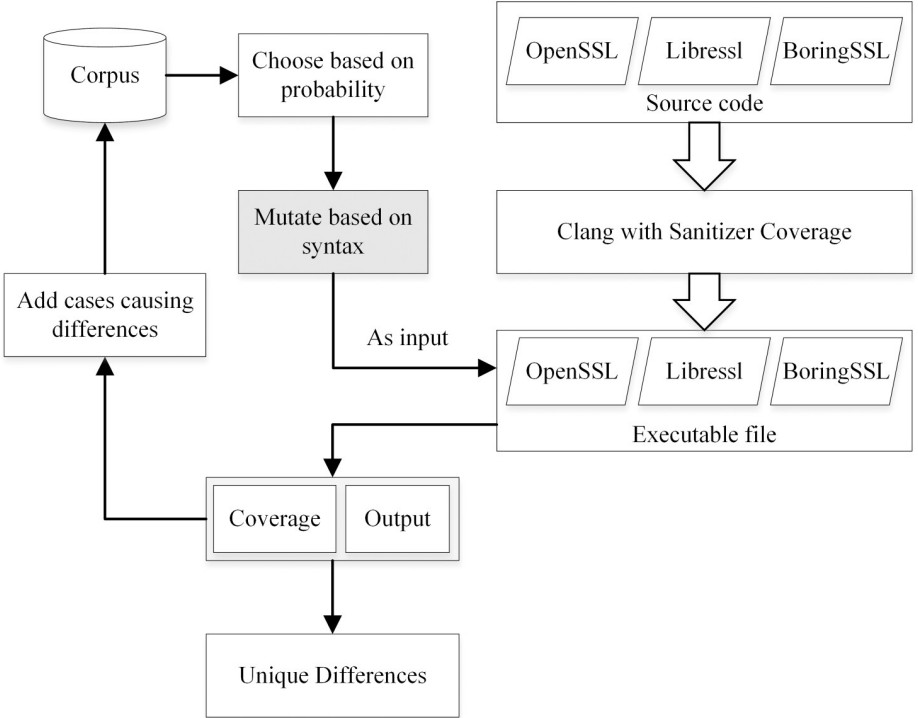

**Fig 1. Integrated framework of the approach.** The two shadow modules are the improvement to the NEZHA.

The abstract function of the response packet used by TLS-diff evaluates the difference in response data,

$$R_1(t) := \begin{cases} 0 & if \ t \equiv ServerHello \\ 21 & otherwise \end{cases} \tag{1}$$

When the response packet is ServerHello, this abstract function is 0; otherwise, it returns 21.

Because TLS-diff is a pure black-box analysis of responses, it cannot record test cases coverage data, making the tool itself unable to use the method of judging discrepancies proposed in this paper. Hence, an appropriate number of test cases were generated as the seeds of the CGDTSM. Since the number of seeds exceeds the maximum number of test cases, the mutation strategy will be invoked, and the result will be the number of discrepancies produced by the test cases generated by TLS-diff. The original seeds are those in [10].

The maximum number of test cases is set to 80,000 and the maximum time is 1000 seconds, under which conditions the results tend to be flat. Five experiments are performed for each method. The average value of the five experiments for each method are taken for comparison.

## Results

To verify the performance of the algorithm, three groups of controlled experiments are designed in this article:

- Three implementations of the latest version were taken, which are hereafter referred to as the new version;

- Three implementations of the version in the original article were taken, which are hereafter referred to as the old version;

- The experiments use three different versions of OpenSSL, called the "SSL-3" version.

It is worth noting that the experiments are based on the same deduplication strategy, although there is still room for improvement on the deduplication strategy. The experimental results are analyzed as follows.

### Q1: How does CGDTSM compare to NEZHA and TLS-diff?

Table 2 is the comparison results of the three methods for the new and old versions. The third line is the average number of discrepancies of five experiments with 1000 seconds. Compared to TLS-diff methods, the number of detected discrepancies found has increased by about 32% and 10% respectively. Fig 2 shows the trend of the number of discrepancies with respect to time under the same standard. At the beginning of testing, TLS-diff shows a better growth trend. However, thanks to the guidance of coverage, the effect of our method exceeded that of TLS-diff.

Through manual analysis of discrepancies, it turned out that there are still many repetitive test cases. And Tables 3 and 4 represent the results after coarse manual analysis. Table 3 represents the results of a horizontal comparison of the above experiments. For the three experiments "new", "old" and "SSL3", we find 18, 13, and 7 unique discrepancies respectively. Table 4 is the detailed number of discrepancies obtained by testing the "new" version. In the first three columns of the table, 21 means that the protocol alarms a test case, and 0 means the clientHello packet is valid and can be negotiated. The first line means that there is a different test case that makes OpenSSL 3.0.0 and LibreSSL 3.2.1 respond to the handshake packet, and at the same time cause BoringSSL-master to response a warning. The meaning of other lines can be inferred similarly.

In summary, the syntax-based mutation is more appropriate than a random mutation, which provides increased coverage for the same number of test cases. Compared to syntax-based uncontrolled mutation, the proposed method can find the same amount of discrepancies faster. Compared to these two methods, there is a certain improvement in the ability to find differences.

### Q2: How effective is the added judgment clause in reducing the duplication of discrepancies?

Since NEZHA and CGDTSM have feedback, the test cases deduplication affects the feedback. Based on the consideration of the one-factor variable, the experiment adopts the TLS-diff method, and the implementation of the new and old versions is tested separately. In addition, the impact of the same judgment condition on the deduplication of the duplicate discrepancies is analyzed. As the number of test cases increases, the trend toward increasing discrepancies between the new and old versions within the deduplication and origin strategies is shown in Fig 3. Red and blue represent the deduplication and origin, respectively. The dashed line and the solid line represent the implementation of old and new versions,

**Table 2. The average discrepancies of the old and new versions based on the three methods.**

| Version | Old | | | New | | |
|---|---|---|---|---|---|---|
| strategy | NEHA | TLS-diff | CGDTSM | NEHA | TLS-diff | CGDTSM |
| Average discrepancies | 16.0 | 119.8 | 132.2 | 57.4 | 188.2 | 248.2 |

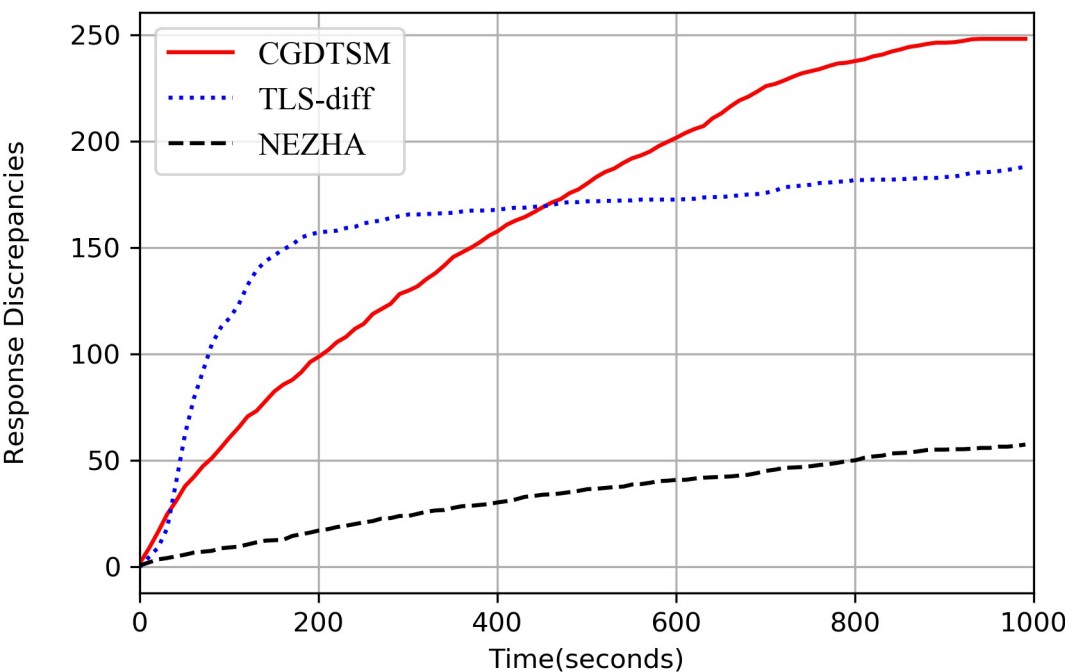

**Fig 2. Comparison of three algorithms based on the "new" version.** Dashed line: NEZHA, dotted line: TLS-diff, solid line: CGDTSM. The three lines represent the response difference obtained by the three methods in 1000 seconds. The experimental object are OpenSSL3.0.0, LibreSSL3.2.1, and BoringSSL(3743aaf).

respectively. The number of response discrepancies after deduplication is significantly lower than that of without deduplication, and the growth trend is slower. With the increasing number of test cases, there are more duplicate discrepancies. The detailed number of using and not using the deduplication strategy are shown in Table 5. Without the deduplication strategy, that is, the strategy used in TLS-diff, we can see that the number of discrepancies produced by the new version is 447.2, which is about 85.4 after deduplication. The number after 81% represents a reduction between the two strategies. In other words, the deduplication strategy reduces repeated use cases by 81%. Similarly, the number of discrepancies produced by the version is 936.8. After deduplication, it is 63.6, indicating that about 93% of repeated use cases can be removed. Thus, the deduplication strategy has a more obvious deduplication effect, which can further reduce the cost of manual analysis.

## Investigation of some discrepancies

The various use cases generated by the tool are analyzed manually. It was found that the differences caused by parsing different implementations in the case the RFC specification are not specified clearly. An analysis of the last three implementations is presented as follows.

**Table 3. The total number of discrepancies for the three experiments.**

| Version | Number |
| --- | --- |
| Openssl3.0.0 vs LibreSSL3.2.1 vs boringssl-master | 18 |
| Openssl1.0.2 vs LibreSSL2.4.0 vs boringssl2883 | 13 |
| Openssl3.0.0 vs Openssl1.1.0 vs Openssl1.0.2 | 7 |

Table 4. The detailed number of discrepancies for the "new" version.

| OpenSSL3.0.0 | LibreSSL3.2.1 | BoringSSL-master | Number |
|---|---|---|---|
| 0 | 0 | 21 | 1 |
| 0 | 21 | 21 | 1 |
| 0 | 21 | 0 | 3 |
| 21 | 0 | 0 | 6 |
| 21 | 0 | 21 | 2 |
| 21 | 21 | 0 | 5 |

## OpenSSL

The parsing of the OpenSSL specifications for Supported Point Formats Extension in RFC4492 is incompatible with BoringSSL and LibreSSL. The definition of ECPointFormat structure in RFC4492 is shown in Table 6, which contains three types of ECPointFormat, and the uncompressed is the type that should be supported. In other words, this type should be enabled, if the ec_point_formats extension exists in the ClientHello package. The lower half of Table 6 is the source code for handling the ECPointFormat in OpenSSL. The implementations of BoringSSL and LibreSSL meet these conditions. It is not validated that clienthello packets do not contain uncompressed type in OpenSSL, which caused the difference.

## LibreSSL

The original maximum length of an SSL record chunk is $2^{14}$. Due to the bandwidth limitation which is caused by Internet of Things devices, the chunk length needs to be adjusted in specific situations. The extension to negotiate maximum fragment length negotiatio n is defined in

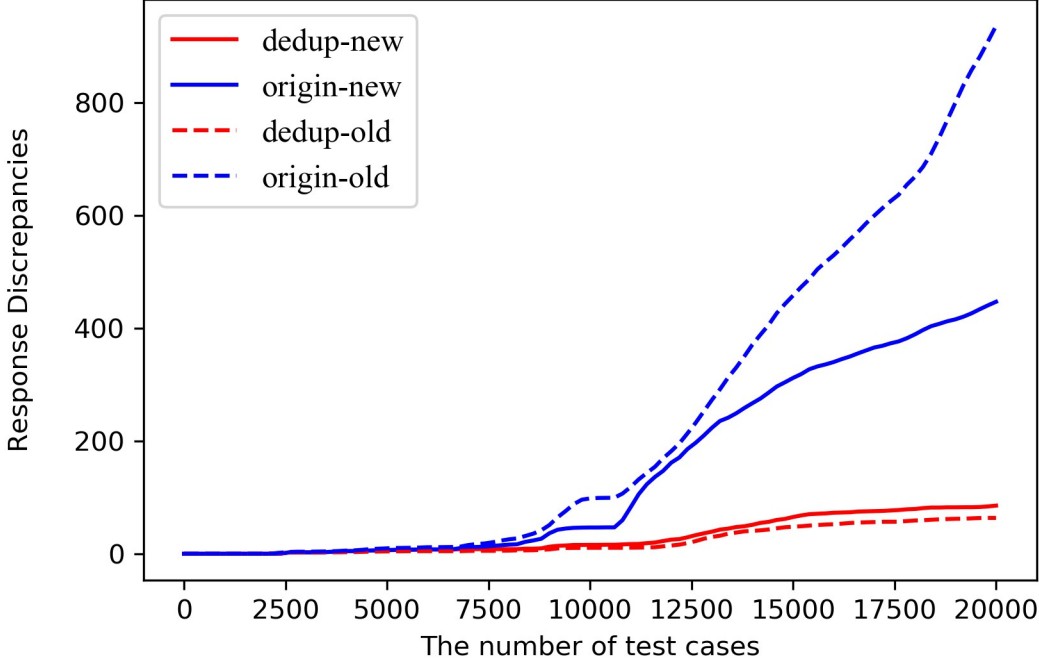

**Fig 3. TLS-diff based deduplication versus no-duplication comparison.** The dashed lines and the solid lines represent the implementation of old and new versions, respectively. The red lines are the number of response discrepancies after deduplication, which are lower than that of without deduplication (blue lines).

**Table 5. The effect of the deduplication strategy on old and new versions.**

| Version | Old | | New | |
|---|---|---|---|---|
| strategy | Origin | Dedup | Origin | Dedup |
| Average discrepancies | 936.8 | 63.6(93%) | 447.2 | 85.4(81%) |

RFC6066. The client should include the max_fragment_length extension in the clientHello to negotiate this type, and use the following values: $2^9$(1), $2^{10}$(2), $2^{11}$(3), $2^{12}$(4), (255). The server can respond to this, and if the requested value exceeds the specified range, it must alert "illegal_parameter". In the latest implementation, Libressl still does not support this extension, while OpenSSL and BoringSSL do support this extension. In experiment, if a test case contains outliers in an extension, Openssl and BoringSSL alarms, but LibreSSL does not validate the extension, resulting in inconsistencies.

## BoringSSL

To negotiate the use of Secure Real-time Transport Protocol security policy, RFC5764 emphasizes that the client should include the use_srtp extension, and the server should select one of the profiles sent from the client for subsequent interactions. If the server fails to find a profile available to the client, the server should return an appropriate DTLS warning. As shown in Table 7, if the profiles provided in the test case are empty, then BoringSSL fails to make a decision and does not raise an alarm. Both OpenSSL and Libressl issue judgment and alarm.

In addition, different implementations make inconsistent judgments of the content fields in parsing some extensions. For example, for parsing renegotiation extensions, OpenSSL only

**Table 6. Code for validating the ECPointFormat.**

```
1 enum {
2    uncompressed (0), ansiX962_compressed prime (1),
3    ansiX962_compressed_char2 (2), reserved (248..255)
4 } ECPointFormat;
5 struct {
6   ECPointFormat ec_point_format_list <1..2^8−1>
7 } ECPointFormatList;
8 int tls_parse_ctos_ec_pt_formats()
9 {
10   if (! PACKET_as_length_prefixed_1 (pkt, &ec_point_format_list)
11      || PACKET_remaining (&ec_point_format_list) == 0) {
12      SSLfatal (s,SSL_AD_DECODE_ERROR, SSL_F_TLS_PARSE_CTOS_EC_PT_FORMATS,XXX);
13      return 0;
14   }
15 }
```

**Table 7. Code for parsing negotiates srtp.**

```
1 static bool ext_srtp_parse_clienthello() {
2    const STACK_OF() * server_profiles = SSL_get_srtp_profiles(ssl);
3    for (const * server_profile: server_profiles) {
4       while (CBS_len(&profile_ids_tmp) > 0) {
5          ……
6       }
7    }
8    return true;
9 }
```

parses the length bytes, while LibreSSL makes further judgment about subsequent bytes. OpenSSL performs preliminary content analysis of the online certificate status protocol OCSP (Online Certificate Status Protocol) extension. If the RFC requirements are not met, an error will be reported, while BoringSSL only reads the status_type field and makes no any judgment about the next content.

## Conclusion

This paper proposes a coverage-guided differential testing method based on syntax mutation for TLS, which is mainly focused on handling clientHello packets by TLS, and combines the ideas of syntax mutation and guided testing. Regarding the problem that discrepancies are due to the same reason, the duplicate test case is specifically defined in collection, and the necessary conditions for duplicate discrepancies are given. Accordingly, a deduplication strategy is formulated, and the rationality of differential guidance in the general situation is analyzed, which is expected to expand to the other differential testing experiments, such as certificate verification.

Using Openssl, LibreSSL, and BoringSSL in an experiment, the proposed method is compared with NEZHA and TLS-diff tools to verify the effectiveness of the hybrid method in the process of finding discrepancies. Meanwhile, the adopted deduplication strategy can effectively eliminate about 87% of repeated discrepancies, reducing the cost of manual analysis.

However, manual analysis is still required to figure out the root cause of the difference. Based on the assessment of the existing target, the difference in code coverage is only a necessary condition of different cases caused by different reasons, and the necessary condition is weak. Therefore, there is still a certain amount of repeated use cases in the output result. For further removal of the duplicate discrepancies, we can use various code coverage tools to eliminate them as much as possible, such as gcov (the code coverage statistics tool of GCC). In addition, investigation of necessary and sufficient conditions for duplicate discrepancies, and automatic determination of the location of the code location that causes the difference in the output will be done in the next study. This work can be extended to compare the differences between patches so that the causes of vulnerabilities can be analyzed.

Beyond that, the current guidance is only intended to extract meaningful use cases and use them as seeds for the next mutation. The mutation strategy can be customized by further analyzing the position of the mutation field and the effect of the mutation operation on the coverage rate. The structure-based differential mutation method used in this article can be extended to other protocols and applications, such as DTLS protocol, to analyze the differences between other applications. We are working on the development of this tool. Besides, as discussed in the paper [11], fully interactive differential testing will be the most interesting direction for future work.

## Supporting information

**S1 File.**
(TXT)

**S1 Code.**
(ZIP)

## Author Contributions

**Conceptualization:** Yan Pan.

**Data curation:** Yan Pan.

**Formal analysis:** Yan Pan, Yuefei Zhu.

**Funding acquisition:** Yan Pan.

**Investigation:** Yan Pan.

**Methodology:** Yan Pan.

**Project administration:** Yan Pan, Yuefei Zhu.

**Resources:** Yan Pan.

**Software:** Yan Pan.

**Supervision:** Yan Pan, Wei Lin.

**Validation:** Yan Pan, Yubo He.

**Visualization:** Yan Pan.

**Writing – original draft:** Yan Pan.

**Writing – review & editing:** Yan Pan, Yuefei Zhu.

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
