## [Decision Letter · Decision Letter 0]

1 Nov 2021

PONE-D-21-31321Coverage-guided differential testing of TLS implementations based on syntax mutationPLOS ONE

Dear Dr. Zhu,

Thank you for submitting your manuscript to PLOS ONE. After careful consideration, we feel that it has merit but does not fully meet PLOS ONE’s publication criteria as it currently stands. Therefore, we invite you to submit a revised version of the manuscript that addresses the points raised during the review process.

We look forward to receiving your revised manuscript.

Kind regards,

Licheng Wang

Academic Editor

PLOS ONE

Journal Requirements:

Reviewers' comments:

Reviewer's Responses to Questions

**Comments to the Author**

1. Is the manuscript technically sound, and do the data support the conclusions?

Reviewer #1: Yes

Reviewer #2: Yes

2. Has the statistical analysis been performed appropriately and rigorously? 

Reviewer #1: I Don't Know

Reviewer #2: Yes

3. Have the authors made all data underlying the findings in their manuscript fully available?

Reviewer #1: Yes

Reviewer #2: Yes

4. Is the manuscript presented in an intelligible fashion and written in standard English?

Reviewer #1: Yes

Reviewer #2: Yes

5. Review Comments to the Author

Reviewer #1: This paper offers a solution to efficiently find differences in the TLS protocol handshake process.

The experimental results proved the method is effective.

However, there are some editorial errors, such as

P2 99：

“243,246 certificates were collected over the network, and the set of the Frankencerts certificate 100 was generated by randomly changing the fields in the certificate. ”

In addition, figures should locate above the caption. It took me a lot of effort to find them.

Reviewer #2: In this paper, the authors propose a hybrid methodology to find differences more efficiently by targeting the TLS protocol handshake process, and then help to discovering the vulnerabilities. In addition, they explore the duplicate discrepancies theoretically and give a method of eliminating duplicate discrepancies based on code. By experiment testing on three types of widely used TLS protocol implementations (OpenSSL, BoringSSL, and LibreSSL), it shows that their work can effectively improve the ability to find differences between different implementations. Thus I suggest accept it after some minor modifies.

1. In the article, there are many abbreviations which make reading difficult, so I suggest the authors give an explanation when they first appear, or list a table to explain them.

2. The presentation quality is to be polished. For example,

--line 18 in the Abstract, “the proposed method can effectively improve finding differences between…”: improve ability or speed ??

--line 41-42 on page 2, “the mutation…due to the….and due to the… ”: please delete the second “due to”.

--line 176 on page 5, “First” should be “Firstly”.

--line 206 on page 6, “The related definitions used in this article are introduced below” should be “The related definitions which are used in this…..”

--line 230:"which is are triggered", line 237: "c1,c2 is a ...??".

--line331-332, “there are 1..cases.. ”

--line 337-338, "Due to the ...cased ..use of ???"

Or I suggest the author to find a professional organization to polish English.

6. PLOS authors have the option to publish the peer review history of their article (what does this mean?). If published, this will include your full peer review and any attached files.

Reviewer #1: No

Reviewer #2: No

---

## [Author Response · Author response to Decision Letter 0]

7 Nov 2021

Dear Editors and Reviewers:

 On behalf of my co-authors, we appreciate editors and reviewers for their constructive comments and suggestions on our manuscript entitled “Coverage-guided differential testing of TLS implementations based on syntax mutation” (PONE-D-21-31321). We tried our best to revise our manuscript according to the comments and described the chagnes in the attachments. We look forward to your response.

---

## [Decision Letter · Decision Letter 1]

19 Dec 2021

Coverage-guided differential testing of TLS implementations based on syntax mutation

PONE-D-21-31321R1

Dear Dr. Zhu,

We’re pleased to inform you that your manuscript has been judged scientifically suitable for publication and will be formally accepted for publication once it meets all outstanding technical requirements.

Kind regards,

Licheng Wang

Academic Editor

PLOS ONE

Additional Editor Comments (optional):

Based on two round reviewing process, including your responses and revising efforts, we would like to tell you that this paper can be accepted for publication.

Reviewers' comments:

Reviewer's Responses to Questions

**Comments to the Author**

1. If the authors have adequately addressed your comments raised in a previous round of review and you feel that this manuscript is now acceptable for publication, you may indicate that here to bypass the “Comments to the Author” section, enter your conflict of interest statement in the “Confidential to Editor” section, and submit your "Accept" recommendation.

Reviewer #1: All comments have been addressed

Reviewer #2: (No Response)

2. Is the manuscript technically sound, and do the data support the conclusions?

Reviewer #1: Yes

Reviewer #2: Yes

3. Has the statistical analysis been performed appropriately and rigorously? 

Reviewer #1: Yes

Reviewer #2: I Don't Know

4. Have the authors made all data underlying the findings in their manuscript fully available?

Reviewer #1: Yes

Reviewer #2: (No Response)

5. Is the manuscript presented in an intelligible fashion and written in standard English?

Reviewer #1: Yes

Reviewer #2: (No Response)

6. Review Comments to the Author

Reviewer #1: The experimental results show that the proposed method can effectively improve the ability to find differences between different implementations. However, the illustration of Table 4 made me confused. I hope it will be improved before publishing.

Reviewer #2: In this paper, the authors proposed a method which can effectively improve the ability to find differences between different implementations. And by experiment testing on three types of widely used TLS protocol implementations (OpenSSL, BoringSSL, and LibreSSL), their experimental results also support the ability to find differences between different implementations.

In the revised paper, the authors made a good modification according to the comments of the reviewer, so I would like to recommend acceptance.

7. PLOS authors have the option to publish the peer review history of their article (what does this mean?). If published, this will include your full peer review and any attached files.

Reviewer #1: No

Reviewer #2: No

---

## [Editor Report · Acceptance letter]

11 Jan 2022

PONE-D-21-31321R1 

Coverage-guided differential testing of TLS implementations based on syntax mutation 

Dear Dr. Zhu:

I'm pleased to inform you that your manuscript has been deemed suitable for publication in PLOS ONE. Congratulations! Your manuscript is now with our production department. 

Kind regards, 

on behalf of

Professor Licheng Wang 

Academic Editor

PLOS ONE